# Impact of Social Media Use on Mental Health within Adolescent and Student Populations during COVID-19 Pandemic: Review

**DOI:** 10.3390/ijerph20043392

**Published:** 2023-02-15

**Authors:** Marija Draženović, Tea Vukušić Rukavina, Lovela Machala Poplašen

**Affiliations:** 1Leadership and Management of Health Services, School of Medicine, University of Zagreb, 10000 Zagreb, Croatia; 2Andrija Štampar School of Public Health, School of Medicine, University of Zagreb, 10000 Zagreb, Croatia

**Keywords:** COVID-19 pandemic, mental health, internet, social media, adolescent, student

## Abstract

The COVID-19 pandemic has drastically changed our lives. By increased screen time during the pandemic, social media (SM) could have significantly impacted adolescents’ and students’ mental health (MH). This literature review aims to synthesize the research on the impact of SM usage on MH of adolescents and students during the first year of the COVID-19 pandemic. A review of the published literature was conducted in April 2021, through a search of PubMed and Web of Science Core Collection databases. The search yielded 1136 records, with 13 articles selected for this review. Most of the included studies observed the negative impact of SM use on MH of adolescents and students, most noticeably observed were anxiety, depression and stress. More active and prolonged SM usage was associated with a negative impact on MH of adolescents and students. Two studies recorded some potentially positive effects, such as support in coping and providing a sense of connection for those who were isolated due to social distancing measures. Since this review focuses on the early period of the pandemic, future studies should investigate the long-term impact of SM use on adolescents and students MH, with all relevant elements that can enable adequate public health response.

## 1. Introduction

The COVID-19 pandemic is not the first pandemic in the history of humanity, but it is undoubtedly the most severe one since the influenza pandemic in 1918. It has led to unprecedented mitigation efforts that disrupted the daily lives of most of the world’s population.

Beyond the general health repercussions of the pandemic itself, these mitigation mandates, including school closures and widespread lockdowns, combined with economic instability, fear of infection and uncertainty for the future, also present a challenge to the mental health (MH) of many [1]. In particular, this might affect adolescents and students, who highly rely on social contact with their peers [2].

MH is most affected by internal and external stressors during adolescence. The effect of stress in adolescents is exacerbated when accompanied by other stressors, such as the lack of sufficient internal or external resources or poorly developed coping skills [3]. Being deprived of social contacts and forced to adjust to online education, while going through a critical developmental stage, adolescents and students might suffer more severe effects of the COVID-19 pandemic-related stressors than the general population [4].

As reported by UNESCO, at its peak, the pandemic had a significant worldwide impact on the lives of more than 1.6 billion students [5]. In China, nearly 40.4% of the sampled youth were prone to psychological problems, and 14.4% suffered from post-traumatic stress disorder (PTSD) symptoms [6]. Social media (SM) has been gaining an increasingly prominent role in adolescents’ lives in recent decades, especially in recent years. In 2018, 45% of teens said they use the internet “almost constantly”, a figure nearly doubling from the 24% in the 2014–2015 survey. An additional 44% said they go online several times a day, indicating that roughly nine-in-ten teens go online at least multiple times per day [7].

SM has become an increasingly important part of adolescents’ daily lives [7] and the COVID-19 pandemic has further accelerated this trend [8,9]. Many adolescents and students have turned to SM to stay connected with their friends and peers and access information and entertainment during a time when in-person interactions have been greatly restricted [10,11].

SM provides data on the pandemics, but also makes available lots of misinformation. The positive impact of SM during the lockdown is the provision of valuable means for social contact. Still, it can also cause poorer sleep quality, lower self-esteem and higher levels of anxiety and depression [12]. US-based research [7] investigating the impact of SM on teen lives, found that a plurality of teens (45%) believe SM has neither positive nor negative effect on people of their age. Roughly three-in-ten teens (31%) say SM impacts mostly positively while the remaining 24% describe its effect as mostly negative. The most significant positive impact of SM use is connecting with friends and family [7]. The study by Coyne et al. [13] found that the time spent using SM was unrelated to individual changes in depression or anxiety. Contrariwise, a study by O’Reilly et al. [14] observing adolescents between 11–18 years suggested that adolescents perceived SM as a threat to mental well-being. SM can provide a sense of connection using technology to connect and support those isolated or feeling isolated due to physical social distancing measures [15]. It can also be a useful tool for staying informed about the latest developments related to the pandemic and accessing resources and support [16]. In contrast, the main negative impacts include bullying/rumor spreading, harm to relationships due to lack of individual contact, unrealistic views of others’ lives and the onset of distraction/addiction [7].

This study aims to synthesize the existing research on the impact of SM use during the first year of the COVID-19 pandemic, related to the MH of adolescents and students. The following questions guided our inquiry: Does SM use during the first year of the COVID-19 pandemic have a predominantly positive or negative impact on MH within the adolescent and student population? Which MH components have been impacted the most by SM use during the first year of the COVID-19 pandemic within the adolescent and student population?

## 2. Materials and Methods

### 2.1. Design

This literature review was conducted in accordance with the guidelines for the preferred reporting items for systematic reviews and meta-analyses [17], with minor modifications where appropriate.

The need to assess the impact of SM use during the first year of the COVID-19 pandemic on MH of adolescents and students is an important health issue. The narrative qualitative synthesis was undertaken with the guidance of the PRISMA (Preferred Reporting Items for Systematic Reviews and Meta-Analyses) 2020 Statement [18].

### 2.2. Search Strategy

A literature search was performed on 30 April 2021 using two databases, PubMed and Web of Science Core Collection.

The searches were conducted using the following defined search terms: (“covid 19” [MeSH Terms] OR “covid 19 vaccines” [MeSH Terms] OR “covid 19 serotherapy” [All Fields] OR “covid 19 nucleic acid testing” [MeSH Terms] OR “covid 19 serological testing” [MeSH Terms] OR “covid 19 testing” [MeSH Terms] OR “sars cov 2” [MeSH Terms] OR “severe acute respiratory syndrome coronavirus 2” [All Fields] OR “ncov” [All Fields] OR “2019 ncov” [All Fields] OR “coronavirus” [MeSH Terms] OR “cov” [All Fields]) AND (“social media” [MeSH Terms] OR “social networking” [MeSH Terms] OR “twitter” [All Fields] OR “youtube” [All Fields] OR “WeChat” [All Fields] OR “Sina” [All Fields]) AND (“mental health” [MeSH Terms]). The search strategy was limited to studies published in English. The full search strategy used for each database has been included in Appendix A.

### 2.3. Study Inclusion and Exclusion Criteria

Studies were included in this review if they were original research focused primarily on the impact of SM use during the first year of the COVID-19 pandemic, related to the MH of adolescents and students.

Studies were excluded from this review if they were not in English; were not original primary research: reviews, reports, abstracts only, case studies, letters, opinions, commentaries, policies, guidelines or recommendations; did not focus primarily on the SM use effect on MH of adolescents or students; and if SM posts were used for content analysis, which was not focused on MH issues.

### 2.4. Data Collection Process and Extraction

Following the search, conducted by an information retrieval specialist (LMP), all references captured by the search engine were uploaded into the reference management software Zotero 6.2 (the Corporation for Digital Scholarship, Virginia, and the USA). Duplicates were identified and removed by MD. The remaining references were uploaded into the Rayyan collaborative tool [19]. Rayyan is a web application and mobile app for systematic reviews. It eases the process of the initial screening of abstracts and titles and helps researchers save time when they share and compare include-exclude decisions.

Initial screening was done by two researchers (MD and TVR) limiting results to those that complied with eligibility criteria. Full texts of 25 papers were assessed for eligibility in detail against the inclusion and exclusion criteria for the review. Thus, a total of 13 studies were finally included in this review. Any disagreements between the reviewers at each stage of the study selection process were resolved through discussion.

One author (MD) used a standardized form developed by the research team to extract the details of the included studies. Data were extracted from each study, including: (1) the first author and year of publication, (2) the study title, (3) the country of origin, (4) the study objective, (5) the study design, (6) the study method/sampling, (7) sample characteristics, (8) mental issues observed, (9) positive vs. negative SM impact on MH observed and (10) main results and conclusions relevant to the impact of SM use on MH of adolescents or students. A second author (TVR) verified the extracted information and checked for accuracy and completeness. Differences were resolved through discussion. The agreed evidence was then synthesized narratively.

### 2.5. Assessment of Risk of Bias

The risk of bias was graded according to the JBI Critical Appraisal tool, “Checklist for Analytical Cross-sectional Studies” and “Checklists for Cohort Studies” [20] by one experienced reviewer (TVR). The evaluation was based on answers to 8 questions (yes, no, unclear or not applicable, for cross-sectional studies) or answers to 11 questions (yes, no, unclear or not applicable, for cohort studies). The studies were classified as having low (>70%), moderate (40–70%) or high (<40%) risk of bias.

### 2.6. Data Synthesis

Data were analyzed according to the study outcomes and objectives. Descriptive (narrative) analyses of the included studies were conducted. A narrative synthesis was undertaken with the guidance of the PRISMA (Preferred Reporting Items for Systematic Reviews and Meta-analyses) 2020 Statement [18]. A narrative synthesis accompanies the tabulated results from the study characteristics and describes: how the results relate to the review’s objective and questions; did SM use during the first year of the COVID-19 pandemic have a predominantly positive or negative impact on MH within the adolescent and student population; and what are the recognized positive and negative impacts.

## 3. Results

### 3.1. Search Results

The literature search retrieved 1136 records (641 from PubMed, 495 from Web of Science Core Collection) and after removing duplicates 806 titles and abstracts were screened. Following title and abstract screening, a further 781 articles were excluded leaving 25 to be screened by full text. Twelve articles did not meet the eligibility criteria following full text screening. Thus, a total of 13 studies were finally included in this review [21,22,23,24,25,26,27,28,29,30,31,32,33].

The PRISMA flow diagram of the study selection and review process is displayed in Figure 1.

### 3.2. Methodological Characteristics of the Studies

Characteristics of the included studies are shown in Table 1. Out of 13 studies included in this review [21,22,23,24,25,26,27,28,29,30,31,32,33], all are observational and the majority (11) [21,22,23,24,25,26,27,28,29,31,33] are cross-sectional. Out of a total of 11,975 participants, 584 were observed within two longitudinal studies [30,32].

The total number of participants in these 13 studies was 11,975, ranging from 49 [32] to 2449 [24] participants in a single study.

Out of 13 eligible studies, 3 originated from the USA (observing a total of 1235 participants) [27,31,32], 2 originated from China (observing a total of 1047 participants) [23,30], 2 from Belgium (observing a total of 4173 participants) [26,28] and 1 originated from each: Bangladesh [21], Turkey [22], Japan [24], Pakistan [25], Canada [29] and Palestine [33], providing a relatively representative sample of student/adolescent population from North America [27,29,31,32], Asia [21,22,23,24,25,30,33] and Northwestern Europe [26,28] including the countries on various levels of development and wealth.

Participants in the studies were described as adolescents, elementary, high school or university students. Therefore, the participants ranged from 6 [33] to 48 [27] years old. The mean (in some cases average) age of participants (with the exclusion of studies where such data is not available) spans from 10.32 [30] to 22.92 [27] years old. Therefore, a limited number of adults [27] among the participants included in some studies is not considered a population with a significant impact on the study results. Additionally, we observed that the majority of participants in most of the studies were female [22,23,25,26,27,28,29,30,31,32,33] (even up to 87.6%) [27].

Most studies used online questionnaires or surveys [21,22,23,24,25,26,27,28,29,30,31,32,33], recruiting participants through the internet or social networks [21,23,24,26,28,29,31,32,33]. In some cases, the participants were recruited through institutional e-mails [24], via school [26,30], national news outlets [28] or within specific classes [27]. One longitudinal study [32] used individual-level online data (Google Search and YouTube) to analyze and correlate to the data collected via questionnaires before and during the pandemic.

Online questionnaires were generally organized in sections, with the first section collecting primary data on students and the following sections collecting specific data necessary for measuring behaviors that might impact MH and indicators of the MH condition. Some studies used a variety of original or somewhat modified validated questionnaires and scales (e.g., PHQ-9 [21,24,31,32], GAD-7 [21,26,31,32], RULS [26,28,29]) and DASS-21 [27,30]), while in the majority of the studies questionnaires and the measures used in them were a combination of validated and self-developed instruments [22,23,24,25,27,28,29,32]. One study used a questionnaire for which the validation status could not be established [33].

### 3.3. Objectives and Outcomes of Included Studies

The objectives and outcomes of included studies are shown in Table 2. Several MH disorders or issues have been observed and measured in the papers. Depression [21,24,27,29,31,32], anxiety [22,26,27,29,31,32] and stress [23,27,29] were listed in three or more studies, while panic [33], addictive SM use [23], mental and psychological distress [28,30], worsening of sleep pattern, lack of motivation and family arguments [25] were connected to SM use in two or fewer studies. Additionally, a nonspecific MH disorder—MH imbalance [21]—was linked to SM use.

Active and increased, and daily use of SM was associated with an increased risk of depressive [24,27,29,31], anxiety [31] and stress [27] symptoms. Additionally, individuals with increasing anxiety or depressive disorders during the pandemic usually had longer sessions using SM [32].

The interaction between COVID-19 stress and SM use was also significant [29]. Individuals suffering more COVID-19 stress had an increased risk of addictive SM use, which has been fostered by active use and flow experience [23].

A significant positive statistical correlation was found between SM and spreading panic concerning COVID-19 [33].

Time spent on SM explained problematic SM use, and problematic SM use subsequently explained psychological/mental distress [30] with odds of psychological/mental distress 3-fold greater for those with an increase in SM use for more than three hours [28].

This review has found that most reviewed papers report predominantly negative impacts of SM use in the COVID-19 pandemic on MH of adolescents [21,22,23,24,27,28,29,30,31,32,33].

Several reviewed studies revealed that increased SM use was related to the MH disorders of students, such as depression, anxiety and stress [22,23,24,27,28,29,31,32]. It also correlated with tiredness, lack of motivation and negative impact on family arguments [25]. Such increased SM use was found to be connected to problematic [30] and addictive SM use [23], potentially leading to mental distress [28,30] or MH imbalance [21], and interacting with COVID-19-related stress [23,29]. College belongingness, which influenced student psychological adjustment, was found to be moderated by SM addiction [22].

Two studies, however, indicated some potentially positive influences of SM on MH, such as long periods of sleep [25] and support in coping through humoristic content and positive exchange in SM [26].

Results of a few studies highlighted the gender difference, indicating that more women than men were found to experience significant mental distress [28,29].

Among the negative impacts of increased or problematic SM use on the MH of adolescents and students most noticeably observed are depression [24,27,29,31,32], stress [23,27,29] and anxiety [22,31,32].

### 3.4. Risk of Bias

The risk of bias in 85% (11/13) of the included studies was classified as low [21,24,26,27,28,29,31,32,33], according to the JBI Critical Appraisal tools [20], as presented in Table 3a,b. In total, only two studies showed a moderate risk of bias [22,25].

## 4. Discussion

### 4.1. Principal Findings

A significant impact of SM on the lives of adolescents and students was evident even before the COVID-19 pandemic and it resulted in both positive and negative outcomes [2,3,14,34]. Some previous studies indicated that the influence of SM use on MH of adolescents might be mostly neutral, even for adolescents suffering from depression and anxiety [7,13].

The studies included in this review originate from multiple countries, providing a sample of the student/adolescent population from North America, Asia and Europe, thus including countries on various levels of development and wealth.

According to this literature review, the influence of SM use on the MH of adolescents and students during the COVID-19 pandemic has been significant. The findings of this review indicate that SM use was predominantly associated with the mental ill-being of adolescents and students during the early months of the COVID-19 pandemic [21,22,23,24,25,27,28,29,30,31,32,33], most commonly related to MH problems, such as depression, anxiety and stress [21,22,23,24,27,28,29,31,32], which is in line with recent publications regarding SM use and its influence on MH of the younger population during the COVID-19 pandemic [35,36,37,38,39,40,41].

Among the articles reviewed in our study, seven studies investigated association between SM use and stress [21,23,27,28,29,30,31], but the term stress was used inconsistently. It was presented as stress in general [27,31], COVID related stress [23,29] or within constructs of mental distress [28], psychological distress [30] or MH imbalance [21].

Two studied stress described as COVID-19 related, either as stressful events [30] or reported stress associated with the initial COVID-19 crisis [29].

Zhao et al. [23] assessed participants experience of COVID-19 related stressful events. COVID-19 stress was significantly positively correlated with active use, SM flow and addictive SM use. Ellis et al. [29] assessed COVID-19 stress, using an adopted version of the Swine Flu Anxiety scale. Items were designed to assess fear about the spread of COVID-19 and the possibility of being infected and specific adolescents concerns that may result from physical distancing. They have also assessed depression (using six-item depression subscale of the Brief Symptom Inventory—BSI) and measured participant loneliness (using the revised UCLA Loneliness Scale—RULS). COVID-19 stress was a significant predictor of depression. The interaction between COVID-19 stress and SM use was also significant. The analysis revealed that the relationship between COVID-19 stress and depression was strongest among adolescents who reported the highest SM use after the pandemic as compared to adolescents with lower and average use (*p* < 0.001).

Three studies assessed psychological distress, but under different terms, as mental distress [28], psychological distress [30] or MH imbalance [21].

Rens et al. [28] used GHQ-12 for the assessment of mental distress. Their results indicate experiencing mental distress were significantly higher among those with small or large increase in SM use. Chen et al. [30] investigated the changes in time spent on use of internet-related activities, changes in problematic use of internet-related activities and changes in psychological distress before and during the school suspension period due to the COVID-19 outbreak. Using 21 items embedded within three subscales of depression (seven items), anxiety (seven items) and stress (seven items), the DASS-21, they have assessed psychological distress. According to their results, increased and problematic SM use is significantly associated with psychological distress. Alam et al. [21] measured stress level using Perceived Stress Scale (PSS), but they also assessed depression (using PHQ-9) and anxiety (using GAD-7). They use the term MH imbalance, which was constructed and categorized in four categories, using cluster analysis combination among three MH scales (PSS, GAD-7 and PHQ-9). Students were categorized into four categories of MH imbalance, where 4.32% had mild, 72.7% had moderate, 12.57% had moderately severe and 10.41% suffered from severe MH imbalance. Since psychological distress refers to non-specific symptoms of stress, anxiety and depression [42], the term MH imbalance used in Alam et al.’s study [21] can be presented also as psychological distress. Their results showed that students spending more time on SM (22.60%) were more likely to be severely depressed, anxious and stressed, or as they stated “in severe MH imbalance”.

Depression Anxiety Stress Scale 21 (DASS-21) was also used by Wheaton et al. [27]. Their results indicate that hours per day of SM use weakly yet significantly related to concern about COVID-19 that are linked to stress and depression, but not anxiety and OCD. In this study terms psychological or mental distress were not used.

Murata et al. [31] assessed depression symptoms (using PHQ-9), anxiety symptoms (using GAD-7), PTSD symptoms (using PC-PTSD-5), perceived stress (using Perceived Stress Scale—PSS), lifetime suicidal ideation and behavior (using SITBI) and prolonged grief reactions (using ICG-RC). According to the findings of this study, adolescents were significantly more likely to report clinically significant depression, anxiety and PTSD symptoms, suicidal ideation or behavior, perceived stress and sleep problems compared to adults. Adolescents with more hours spent on SM were more likely to have moderate to severe depressive and anxiety symptoms.

This review found a link between increased SM use and depression [24,27,29,31,32], which is consistent with the findings in recent research where SM exposure [38] and excessive SM networking site usage [39,40] were associated with increased depression. Research has shown that the more time adolescents and students spend on SM, the more likely they are to experience negative effects on their MH [35,36,37,38,39,40,41,42,43], that excessive use of SM can contribute to feelings of loneliness [39], anxiety [36,40] and depression [36,38,39,40]. This is particularly true for those who compare themselves to others on SM [44] and experience cyberbullying [45,46].

Results were similar when looking at anxiety. This review found that adolescents with more hours spent on SM were more likely to have moderate to severe anxiety symptoms [31]. Similarly, individuals with increasing anxiety symptoms during the pandemic usually conduct longer sessions when engaging with SM (YouTube) [32].

Recent research confirms this and finds that anxiety scores were higher in those who used the SM for more than 7 h per day, compared to those who used it for 0–2 or 3–4 h [36] and that excessive time spent on SM platform was associated with a greater likelihood of having anxiety symptoms [40].

Other research also shows that the COVID-19 pandemic has exacerbated existing MH problems among adolescents [47] and SM may exacerbate these problems [48]. For example, the constant stream of news and information about the pandemic on SM can lead to increased levels of stress and anxiety [49]. Additionally, the lack of in-person social support and the increased reliance on SM for social interaction may contribute to feelings of loneliness [34]. There is also some evidence to suggest that SM use may interfere with sleep quality and quantity among adolescents and students, which can negatively affect their overall MH and well-being [50,51].

Even though the majority of the studies in this review associate increased or problematic use of SM with a predominantly negative impact on the MH of adolescents and students during the early months of the COVID-19 pandemic, two studies, however, indicated some potentially positive influences of SM on MH, such as long periods of sleep [25] and support in coping through humoristic content and positive exchange in SM [26]. A similar beneficial effect of SM use was also observed by other studies, which found that SM can provide a sense of connection and support for those who are isolated or feeling isolated due to social distancing measures [43] or SM was observed to offer a helpful way of educating and reaching adolescents to promote mental well-being and cope with emotional burdens [52,53]. Additionally, other publications found SM useful in providing information about MH [43,53] and substituting live social contacts [54].

Contrarily, SM was used by some to seek support for suicidal thoughts and self-harm [36] and also contributed to poor MH through validation-seeking practices, fear of judgment, body comparison, addiction and cyberbullying [43]. A result from a longitudinal study conducted in Sweden, with a 2-year long follow-up, suggests that increased use of SM might be an indicator, rather than a risk factor for MH symptoms [55].

### 4.2. Limitations

There are several limitations to this review. The search for this literature review was performed in April 2021 using two databases, PubMed and Web of Science Core Collection. Future searches should be optimized by searching additional multi-disciplinary databases, such as Scopus, CINAHL or PsycINFO. The search for reference lists and citations would also be welcomed in the subsequent literature reviews. Only English language articles, presenting original research in a defined period were included; papers in other languages and outside the timeframe for inclusion may have identified additional relevant studies.

This review was conducted according to the guidelines for the preferred reporting items for systematic reviews and meta-analyses [17,18,20], with minor modifications. Even though there are recommendations from the JBI that COVID-19 related reviews should, besides the comprehensive literature of multiple bibliographic databases search (e.g., MEDLINE and WoS), include a search of the gray literature and/or scanning of the references [56]; we have not performed a search of the gray literature nor scanned the references of our final sample. Searching these sources is complex because of a lack of indexing and poor functionality of the search interfaces, thus we omitted it.

The data processed in the studies that were collected were obtained from November 2019 [30] until August 2020 [33] and generally related to the first year of the lockdown. Such data represents the short-term impacts of SM use on the MH of adolescents and students. The limitations imposed on the population due to the outbreak of the COVID-19 pandemic have, however, already lasted much longer than initially expected more than two years. Therefore, the findings of this review are relevant just for the relatively short period at the beginning of the pandemic, the first 16 months of the COVID pandemic, which limits their relevance. However, this period was significant since the most severe lockdown measures were introduced globally, allowing us to review studies from that period, from a specific perspective on the impact of SM use on MH within the most vulnerable populations (adolescents and students). At the same time, the long-term impacts of SM use on the MH of adolescents and students might significantly differ from the shorter-term impacts included in the reviewed papers.

All of the studies were observational and the majority [21,22,23,24,25,26,27,28,29,31,33] were cross-sectional and went no further than describing a prevalence of the specific MH condition. Another limitation was that just two longitudinal studies [30,32] investigating this review’s aims could be found—limiting the time component of reviewed studies. This is probably a consequence of the data collection period occurring relatively early after the pandemic outbreak, so there was a limited opportunity for multiple subsequent research waves on the relevant population samples to be performed.

This notion is also proven by the fact that the included studies used convenient sampling, recruiting participants predominantly via SM and Internet, using online questionnaires. Those methods were the most convenient, practical and feasible methods during the lockdown. Therefore, the results of this review are based on data from the studies, dominantly based on a convenient sample.

Regarding the risk of bias, and quality of the studies in this review’s final sample, only four studies, exactly stated their response rate [24,30,32,33], ranging from 53% to 100%. Even though the risk of bias vas very low in 11 of 13 studies, it is important that future studies report response rates more often to increase the studies’ quality.

The age of the participants spanning from childhood (elementary school students) to adulthood, makes the review population somewhat heterogeneous. However, the mean/average age of participants, ranging from 10.32 [30] to 22.92 [27] years, makes the data used in this study relevant for the population of adolescents and students.

COVID-19 pandemic mitigation efforts have lasted much longer than the period examined in this review. Impacts of SM use during pandemics on the MH of adolescents and students in such a prolonged period might significantly differ from those observed in reviewed papers. Therefore, findings from more recent studies investigating the long-term impact of SM on adolescents and students during the COVID-19 pandemic should also be examined to identify possible differences with outcomes observed in this review.

## 5. Conclusions

Based on the findings of reviewed studies, we conclude that increased or problematic use of SM predominantly negatively impacted the MH of adolescents and students during the first year of the COVID-19 pandemic. The majority of the included studies observed the negative impact of SM on MH, while just two studies recorded some potentially positive effects, such as support in coping and providing a sense of connection for those who were isolated due to social distancing measures. Among the negative consequences of increased or problematic SM use on MH of adolescents and students, most noticeably observed were anxiety, depression and stress. Since this review focuses on the early period of the pandemic, at this point, we can only speculate about the long-term impacts of SM on MH of adolescents and students during the COVID-19 pandemic.

Future studies, especially longitudinal and studies observing the influence of different types of SM behavior and activities, could provide valuable insights and directions for dealing with the influence of SM on the MH of adolescents and students during pandemics since we are clearly facing a new pandemic—an increase of MH disorders among our youngest generations. We should be prepared for how MH care should change due to the COVID-19 pandemic and adequately respond, especially concerning MH of adolescents and students.

## Figures and Tables

**Figure 1 ijerph-20-03392-f001:**
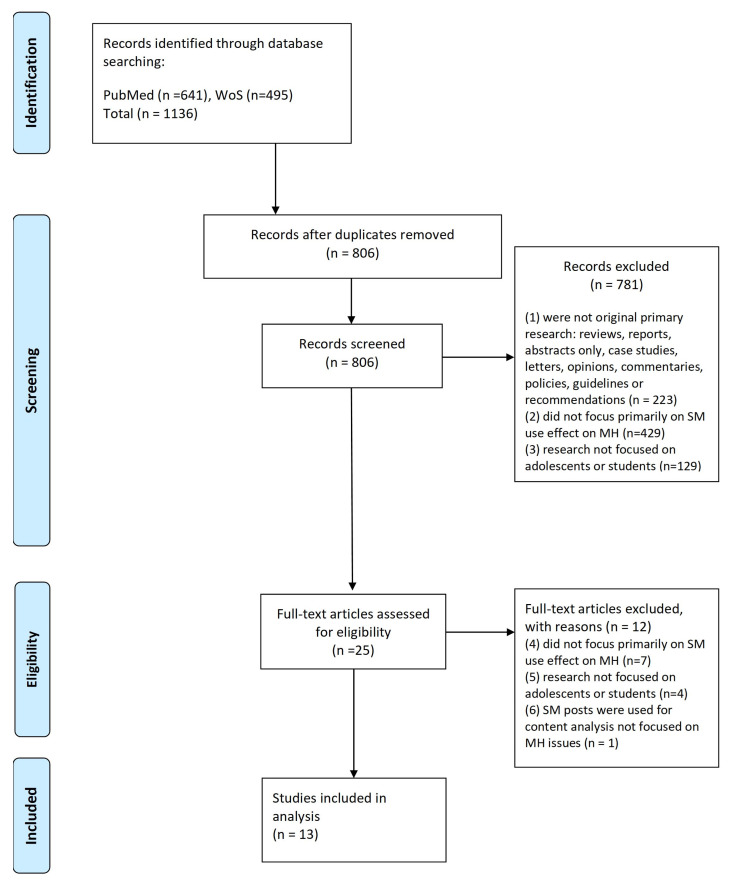
PRISMA (Preferred Reporting Items for Systematic Reviews and Meta-analyses) flow diagram of the study selection and review process.

**Table 1 ijerph-20-03392-t001:** Methodological Characteristics of Included Studies.

Authors	Study Title	Country	Design	Method/Sampling	Participant/Sample Characteristics
Alam, MK et al. (2021) [21]	Assessing the mental health condition of home-confined university level students of Bangladesh due to the COVID-19 pandemic	Bangladesh	Observational, cross-sectional	Online-based questionnaire Distribution: SM Convenient sampling	509 university students of Bangladesh, Age 18–28 yrs. 41.5% female
Arslan, G et al. (2021) [22]	Coronavirus anxiety and psychological adjustment in college students: exploring the role of college belongingness and social media addiction	Turkey	Observational, cross-sectional	Online-based questionnaire Distribution: not specified Convenient sampling	315 undergraduate students, Age 18–39, M = 21.65 ± 3.68 yrs. 67% female
Zhao and Zhou (2021) [23]	COVID-19 stress and addictive social media use (SMU): Mediating role of active use and social media flow	China	Observational, cross-sectional	Online survey Distribution: SM (advertisement on WeChat) Convenient sampling	512 Chinese college students, Age 18–30 M = 22.12 ± 2.47 yrs. 62.5% female
Nomura, K et al. (2021) [24]	Cross-sectional survey of depressive symptoms and suicide-related ideation at a Japanese national university during the COVID-19 stay-home order	Japan	Observational, cross-sectional	Online survey Distribution: institutional e-mails Convenient sampling	2712 (of 5111) Akita university students, RR = 53% Age M = 20 ± 2 yrs. 42% female
Ali, A et al. (2021) [25]	Effects of COVID-19 pandemic and lockdown on lifestyle and mental health of students: A retrospective study from Karachi, Pakistan	Pakistan	Observational, cross-sectional	Online survey Distribution: not specified Convenient sampling with open Epi to calculate sample size	251 students, Age 14–24, average 19.4 yrs. 70.2% female
Cauberghe, V et al. (2021) [26]	How adolescents use social media to cope with feelings of loneliness and anxiety during COVID-19 lockdown	Belgium	Observational, cross-sectional	Online survey Distribution: e-mails via school, organizations and SM Convenient sampling	2165 adolescents, Age 13–19, M = 15.51 ± 1.59 yrs. 66.6% female
Wheaton, MG et al. (2021) [27]	Is fear of COVID-19 contagious? The effects of emotion contagion and social media use on anxiety in response to the Coronavirus pandemic	USA	Observational, cross-sectional	Online survey Sample recruited from psychology classes Convenient sampling	603 psychology classes students, Age 18–48 yrs. M = 22.92 87.6% female
Ellis, WE et al. (2020) [29]	Physically isolated but socially connected: Psychological adjustment and stress among adolescents during the initial COVID-19 crisis	Canada	Observational, cross-sectional	Online survey Distribution: Instagram, e-mail Convenient sampling	1054 high school students, Age 14–18, M = 16.68 ± 0.78 yrs. 76.4% female
Murata, S et al. (2020) [31]	The psychiatric sequelae of the COVID-19 pandemic in adolescents, adults, and health care workers	USA	Observational, cross-sectional	Online survey Distribution: SM and universities Convenient sampling	total participants 4909, adolescents 583, 80% female
Zhang, B et al. (2020) [32]	The relationships of deteriorating depression and anxiety with longitudinal behavioral changes in Google and YouTube use during COVID-19: Observational study	USA	Longitudinal observational	Individual-level online data (Google Search and YouTube) questionnaires prior to and during the pandemic Distribution: digital announcements Convenient sampling	cohort of 49 undergraduate students, RR = 100%, 61% female
Radwan, E et al. (2020) [33]	The role of social media in spreading panic among primary and secondary school students during the COVID-19 pandemic: An online questionnaire study from the Gaza Strip, Palestine	Palestine	Observational, cross-sectional	Online questionnaire Distribution: poster on Virtual Classroom, SM Convenient sampling	985 of 1067 invited students (RR = 92.3%) Age 6–18 yrs. 65.8% female
Chen, IH et al. (2021) [30]	Problematic internet-related behaviors mediate the associations between levels of internet engagement and distress among schoolchildren during COVID-19 lockdown: A longitudinal structural equation modeling study	China	Observational, longitudinal, two waves	Questionnaires, Online survey Distribution: teachers in three schools Convenient sampling	550 school children (1st wave), 535 school children (2nd wave), RR = 98.7% M = 10.32 yrs. 50.5% female
Rens, E et al. (2021) [28]	Mental distress and its contributing factors among young people during the first wave of COVID-19: A Belgian survey study	Belgium	Observational, cross-sectional	Online survey Distribution: SM, national news outlets Convenient sampling	2008 participants Age 16–25 yrs. M = 22.27 ± 2.29 78.09% female

SM—social media; RR—response rate; OpenEpi—free and open-source software for epidemiologic statistics.

**Table 2 ijerph-20-03392-t002:** Objectives and Outcomes of Included Studies.

Authors	Study Objective	Mental Issues Observed/Validated Instruments Used	Positive vs. Negative SM Impact Observed	Main Results and Conclusions
Alam, MK et al. (2021) [21]	To investigate the psychological health challenges faced by Bangladeshi university students during this COVID-19 pandemic.	MH imbalance, depression, anxiety, stress; PHQ-9, GAD-7, PSS.	POS: - NEG: Spending more time on SM and other factors significantly connected with MH imbalances.	The majority of university students suffered from MH disturbances in lockdown. Those using social sites frequently suffered more mental problems than those who used sites once or twice a day.
Arslan, G et al. (2021) [22]	To examine the impact of coronavirus anxiety on psychological adjustment and to explore the mediating and moderating role of college belongingness and SM addiction during the COVID-19 outbreak.	Coronavirus anxiety, lack of psychological adjustment related to a sense of belonging; CAS, CBQ, BSMAS, BASE-6.	POS: - NEG: SM addiction moderated the association between coronavirus anxiety and college belongingness.	SM addiction moderates the association between coronavirus anxiety and college belongingness, which in turn influences student psychological adjustment. Decreasing SM addictive behavior could facilitate college students dealing with coronavirus anxiety and promote their feelings of belongingness, which in turn would improve their adaptive psychological adjustment. College belongingness is a potential mechanism explaining how coronavirus anxiety is related to psychological adjustment and this relation may depend on the levels of SM addiction.
Zhao and Zhou (2021) [23]	To understand the relationships between COVID-19 stress, SM active use, SM flow, and addictive SM use.	COVID stress, addictive SM use; The brief version of BFAS and instruments developed for this study.	POS: - NEG: SM active use mediates relationship COVID stress—addictive SM use.	SM active use, including SM flow, increases addictive SM use. Individuals suffering more COVID-19 stress are at increased risk of addictive SM use that may be fostered by active use and flow experience.
Nomura, K et al. (2021) [24]	To investigate the prevalence of depressive symptoms and suicide-related ideation during the COVID lockdown and provide input for future intervention on depression and suicide prevention.	Depression, suicide-related ideation; Japanese version of the PHQ-9, and instrument developed for this study.	POS: - NEG: Increased risk of depression.	Daily SM communication is associated with an increased risk of depressive symptoms. Negative lifestyles (smoking, drinking), and daily SN communication using either video or voice may be risk factors for depressive symptoms.
Ali, A et al. (2021) [25]	To investigate the correlations between changes in sleep patterns, perception of time flow and digital media usage during the outbreak and the impact of these changes on the mental health of students.	Tiredness, worsened sleep pattern, lack of motivation, family arguments; Instrument developed for this study.	POS: Longer periods of sleep NEG: Increase in tiredness, lack of motivation and family arguments.	An increase in SM usage correlates with tiredness/lack of motivation, and has a negative impact on family arguments. Students who used SM more reportedly slept for longer periods. Increased use of SM led to increased sleep length, worsening sleep habits and a general feeling of tiredness.
Cauberghe, V et al. (2021) [26]	To examine the potential benefit of SM for adolescents coping with feelings of anxiety and loneliness during the quarantine.	Loneliness, anxiety; CESD scale, GAD-7, RULS-6 item, and adopted version of the Brief-coping Scale.	POS: Some SM activities help in actively managing moods and using humor for coping. NEG: -	Using SM as a substitute for physical social relations makes adolescents less happy. SM can be used as an instrument to actively cope with the situation, relieve anxiety, and feel better. Humor on social media is beneficial for adolescents’ well-being during the lockdown. SM can be used as a constructive coping strategy for adolescents to deal with anxiety during the COVID-19 quarantine.
Wheaton, MG et al. (2021) [27]	To investigate the relationship between susceptibility to emotion contagion, media usage and emotional responses to the COVID-19 outbreak.	Depression, anxiety, stress, OCD; DASS-21, OCI-R, ECS, CTS and instrument developed for this study.	POS: - NEG: SM use linked to stress and depression.	Hours per day of SM use weakly yet significantly related to concern about COVID-19 that are linked to stress and depression, not anxiety and OCD. Results showed that media consumption about COVID-19 significantly predicted the degree of COVID-19-related anxiety.
Ellis, WE et al. (2020) [29]	To examine the COVID related stress among adolescents and the relationship between their daily behaviors including SM use, virtual communications with friends, time with family, time completing schoolwork and physical activity on feelings of psychological distress (i.e., depression and loneliness).	Depression, loneliness, COVID stress; Swine Flu Anxiety Scale, BSI, RULS-6 item, Godin Leisure-Time Exercise Questionnaire and instruments developed for this study.	POS: - NEG: Increase in SM use increases depression; significant interaction between COVID-19 stress and SM use.	Greater SM use before and after the COVID-19 crisis was related to higher depression, but not loneliness. COVID-19 stress was related to more loneliness and depression, especially for adolescents who spend more time on social media. For adolescents with depressive symptoms, it may be important to monitor the supportiveness of online relationships.
Murata, S et al. (2020) [31]	To assess COVID pandemics mental health impact across the lifespan in the United States in adolescents, adults and health care workers.	Depression, anxiety, stress, PTSD, suicidal ideation and behavior, prolonged grief reactions; PHQ-9, GAD-7, PC-PTSD-5, SITBI self-report version, ICG-RC, ISI, PSS.	POS: NEG: SM use linked to moderate or severe depression and anxiety.	Adolescents with more hours spent on SM were more likely to have moderate to severe depressive and anxiety symptoms. A pandemic is associated with increased rates of clinically significant psychiatric symptoms, loneliness could put individuals at increased risk for the onset of psychiatric disorders.
Zhang, B et al. (2020) [32]	To examine the relationships of deteriorating depression and anxiety conditions with the changes in user behaviors when engaging with Google Search and YouTube during COVID-19.	Depression, anxiety; PHQ-9, GAD-7 and instruments developed for this study.	POS: - NEG: Correlation between prolonged online activities (YouTube, Google Search) and deteriorated MH.	Results indicate that individuals with increasing anxiety or depressive disorders during the pandemic usually have long use sessions when engaging with Google Search and YouTube. Online behavior significantly correlated with deteriorations in the PHQ-9 scores and GAD-7 scores. Deteriorating depression and anxiety correlate with behavioral changes in Google Search and YouTube use.
Radwan, E et al. (2020) [33]	To determine the effect of SM on the spread of COVID-19 related panic among primary and secondary school students.	Panic; instrument developed for this study.	POS: - NEG: SM spreads panic and has a potential negative impact on MH.	SM has a significant impact on spreading panic and potentially negatively impacting their mental health and psychological well-being. SM has a main role in rapidly spreading panic about the COVID-19 pandemic among students in the Gaza Strip.
Chen, IH et al. (2021) [30]	To (i) assess changes in the level of engagement in three internet-related activities (smartphone use, social media use, and gaming) before and during the COVID-19 outbreak, including prolonged and problematic engagement in these activities; (ii) investigate the differences of psychological distress before and after COVID-19 outbreak; and (iii) to use structural equation modeling to investigate the mediating roles of problematic internet-related behaviors in the causal relationships of psychological distress and time spent on internet-related activities.	Psychological distress; SABAS, BSMAS, IGDS-SF9, DASS-21.	POS: - NEG: Problematic SM use is significantly associated with psychological distress.	Time spent on SM significantly explained problematic SM use, problematic SM use subsequently explained psychological distress. Increased problematic use of internet-related activities among schoolchildren was associated with greater psychological distress.
Rens, E et al. (2021) [28]	To improve understanding of the associated factors of mental distress among 16–25-year-olds during the beginning of the first wave of the COVID-19 pandemic in Belgium	Mental distress; GHQ-12, OSSS-3, an adapted version of RULS-3 item and instruments developed for this study.	POS: - NEG: Increased SM use significantly predicts mental distress.	Mental distress is highest among women, those experiencing loneliness and those whose everyday life is most affected. The psychological needs of young people, such as the need for peer interaction, should be more recognized and supported.

BASE-6—Brief Adjustment Scale-6, BFAS—Bergen Facebook Addiction Scale, BSI—Brief Symptom Inventory, BSMAS—Bergen Social Media Addiction Scale, CAS—Coronavirus Anxiety Scale, CBQ—College Belongingness Questionnaire, CESD—Center for Epidemiologic Studies Depression Scale, CTS—COVID Threat Scale, DASS-21—Depression Anxiety Stress Scales 21, ECS—Emotion Contagion Scale, GAD-7—The General Anxiety Disorder Scale, GHQ-12—General Health Questionnaire, ICG-RC—Inventory for Complicated Grief-Revised for Children, IGDS-SF9—Internet Gaming Disorder Scale-Short Form, ISI—Insomnia Severity Index, MH—mental health, NEG—negative, OCD—obsessive compulsive disorder, OCI-R—Obsessive Compulsive Inventory-Revised, OSSS-3—3-item Oslo Social Support Scale, PC-PTSD-5—Primary Care Post-traumatic Stress Disorder Screen for Diagnostic and Statistical Manual of Mental Disorders-5, PHQ-9—Patient Health Questionnaire-9, POS—positive, PSS—Perceived Stress Scale, RULS-Revised UCLA Loneliness Scale, SABAS—Smartphone Application-Based Addiction Scale, SITBI—Self-Injurious Thoughts and Behaviors Interview, SM—social media, SN—social network.

**Table 3 ijerph-20-03392-t003:** (a) Assessment of Risk of Bias. The Joanna Briggs Institute (JBI) Critical Appraisal tool. Checklist for Analytical Cross- Sectional Studies [20]. (b) Assessment of Risk of Bias. The Joanna Briggs Institute (JBI) Critical Appraisal tool. Checklist for Cohort Studies [20].

**(a)**													
**Studies**	**Were the Criteria for Inclusion in the Sample Clearly Defined?**	**Were the Study Subjects and the Setting** **Described in Detail?**	**Was the Exposure Measured in a Valid and Reliable way?**	**Were Objective, Standard Criteria used for Measurement of the Condition?**	**Were Confounding Factors Identified?**	**Were** **Strategies to Deal with Confounding Factors Stated?**	**Were the Outcomes Measured in a Valid and Reliable way?**	**Was Appropriate Statistical Analysis Used?**				**Total Number of Yes % ***	**Risk of Bias ****
Alam, MK et al. [21]	Yes	Yes	Yes	Yes	Yes	Yes	Yes	Yes				100%	Low
Ali, A et al. [25]	Yes	Yes	Unclear	No	NA	NA	Yes	Yes				67%	Moderate
Arslan, G et al. [22]	No	No	Yes	Yes	NA	NA	Yes	Yes				67%	Moderate
Cauberghe, V et al. [26]	Yes	Yes	Yes	Yes	Yes	Yes	Yes	Yes				100%	Low
Ellis, WE et al. [29]	Yes	Yes	Unclear	Yes	Yes	Yes	Yes	Yes				88%	Low
Murata, S et al. [31]	Yes	Yes	Yes	Yes	Yes	Yes	Yes	Yes				100%	Low
Nomura, K et al. [24]	Yes	Yes	Yes	Yes	Yes	Yes	Yes	Yes				100%	Low
Radwan, E et al. [33]	Yes	Yes	Yes	Yes	Yes	Yes	Yes	Yes				100%	Low
Rens, E et al. [28]	Yes	Yes	Yes	Yes	Yes	Yes	Yes	Yes				100%	Low
Wheaton, MG et al. [27]	Unclear	No	Yes	Yes	Yes	Yes	Yes	Yes				75%	Low
Zhao and Zhou [23]	Yes	Unclear	Yes	Yes	Yes	Yes	Yes	Yes				88%	Low
**(b)**													
**Studies**	**Were the Two Groups Similar and Recruited from the Same Population?**	**Were the Exposures Measured Similarly to Assign People to Both Exposed and Unexposed Groups?**	**Was the Exposure Measured in a Valid and reliable way?**	**Were Confounding Factors Identified?**	**Were** **Strategies to Deal with Confounding Factors Stated?**	**Were the Groups/Participants Free of the Outcome at the Start of the Study (or at the Moment of Exposure)?**	**Were the Outcomes Measured in a Valid and Reliable way?**	**Was the Follow up Time Reported and Sufficient to Be Long Enough for Outcomes to Occur?**	**Was Follow up Complete, and If Not, Were the Reasons to Loss to Follow up Described and Explored?**	**Were Strategies to Address Incomplete Follow up Utilized?**	**Was Appropriate Statistical Analysis Used?**	**Total Number of Yes % ***	**Risk of Bias ****
Chen, IH et al. [30]	Yes	NA	Yes	Yes	Yes	Yes	Yes	Yes	Yes	Yes	Yes	100%	Low
Zhang, B et al. [32]	Yes	NA	Yes	Yes	Yes	NA	Yes	No	Yes	Yes	Yes	89%	Low

* NA = Not Applicable. ** Low risk of bias >70%; Moderate risk of bias 40–70%; High risk of bias < 40%. The percentage was calculated according to have many “yes” each study got relative to the applicable items.

## Data Availability

Data are contained within the article.

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
