# Peer review of "Impact of Social Media Use on Mental Health within Adolescent and Student Populations during COVID-19 Pandemic: Review"

_ijerph, 2023, doi:10.3390/ijerph20043392_

Round 1

Reviewer 1 Report

The topic of this rapid review is somehow meaningful during the COVID-19 pandemic, especially among adoelsents and young adults who were deprived of social interactions with peers and had to adjust to online education. However, there are few concerns and suggestions to be addressed in this manuscript:

--A major concern of this study is the timeliness of the results. The rapid review was performed in April 2021 which was nearly two years ago, and I’m worried that the conclusions from this manuscript would be very out dated. Inclusion of more updated studies is necessary.

--The small sample size seems unpersuasive as a rapid review.

--Since the title of your study is targeting the period of the COVID-19 pandemic, please explain why you included your reference [30] of a longitudinal study with first wave of data collected in November 2019 when COVID-19 did not break out yet.

--More detailed descriptions of the results should be included in the abstract.

Reviewer 2 Report

This work and its results are carefully designed and written. I only would like to raise the authors' attention to some minor inaccuracies:

Introduction:

p1, line 30-36: 'Beyond the general health repercussions of the pandemic itself, these mitigation mandates, including school closures and widespread lockdowns, combined with economic instability, fear of infection, and uncertainty for the future, also present a challenge to the mental health (MH) of many [1]. In particular, this might affect adolescents and students, who highly rely on social contact with their peers [2].

I WOULD START A NEW PARAGRAPH HERE: MH is most affected by internal and external stressors during adolescence. The effect of stress in adolescents is exacerbated when accompanied by other stressors, such as the lack of sufficient internal 36 or external resources, or poorly developed coping skills [3].'  

Materials and Methods

Design: 

p2, line 81-84: 'This rapid review was conducted in accordance with the Joanna Briggs Institute (JBI) position statement about rapid reviews and the methodological rigor of evidence synthesis [17], with minor modifications where appropriate.' THESE MINOR MODIFICATIONS ARE NOT FURTHER CLARIFIED - NEITHER HERE, NOR LATER.

'We chose a rapid review protocol in line with the recommendations by JBI; a rapid review is an approach...' I WOULD DELETE THE FIRST PART OF THE SENTENCE DUE TO REPETITION AND START THE SENTENCE WITH A RAPID REVIEW IS AN APPROACH...

Search strategy:

I do not find Table1 necessary. Why are 'vaccine', 'testing' and 'therapy' among the searched terms? PLEASE, CLARIFY THIS!

Study Exclusion Criteria:

p5, line 114: studies in which 'SM posts were used for content analysis' were excluded. THIS DECISION IS NOT VERY-WELL SUPPORTED, SINCE THIS IS A NARRATIVE REVIEW. DID THE CONTENT-ANALYSIS FOCUS ON MH?

Results:

p7, line 170-171: 'Out of a total of 11,975 participants, just 584 were observed within two 170 longitudinal studies [30,32].' PLEASE, DELETE THE WORD 'JUST', BECAUSE IT IS JUDGEMENTAL!

p8, 177-180: '...originated from each: Bangladesh [21], Turkey [22], Japan [24], Pakistan [25], Canada [29] 177 and Palestine [33], providing a relatively representative sample of student/adolescent population from North America [27,29,31,32], Asia [21–25,30,33] and Europe [26,28] including the countries on various levels of development and wealth.' I do not think Belgium itself is representative for Europe. PLEASE, REFORMULATE THIS PART OF THE SENTENCE! 

Objectives and Outcomes of Included Studies:

p13, 214-215: 'Some nonspecific MH disorders were linked to SM use, e.g. MH imbalance 214 [21], psychological [30] and mental [29] distress.' Distress has several well-articulated definition (e.g. Viertiö et al, 2021): refers to non-specific symptoms of stress, anxiety and depression. PLEASE, CONSIDER TO REFORMULATE THE SENTENCE!

Table3: It would be worthy of interest including the name of all the used, validated questionnaire!

Discussion:

p22, line 282: '...most commonly related to MH problems such as depression, anxiety and stress' PLEASE, REFLECT ON PSYCHOLOGICAL PHENOMENON MORE ACCURATELY: WHAT KIND OF STRESS WAS CITED AND MEASURED HERE? e.g. Acute stress is not necessarily a mental health problem but a reaction to an unknown situation like COVID pandemic.

p22, line 301-302: 'This review found that stress symptoms were associated with SM use [23,27,29], contrary to recent research findings that using SM to obtain health information could reduce stress [49].' I do not think this is a frutfuil comparison: you did not mention what adolescents used SM during the pandemic for (to obtain health information?). Furthermore, quite threatening information could be found about COVID on SM in the first wave of the pandemic . 

Round 2

Reviewer 1 Report

Thanks for the authors responses and efforts in revising this mansucript. However, I think the authors have an unclear understanding about the definition and practical meaning of rapid reviews. 

"Rapid reviews and the methodological rigor of evidence synthesis: a JBI position statement" was used as one of the guidance paper of this rapid review. Timeliness was repeatedly emphasized as an essential character of rapid review throughout out this article. Rapid reviews aim to provide timely information and rapid-turnaround evidence for decision-makers who need guidance in fast-paced contexts, although it may require methodological trade-offs (e.g., omission, abbreviation, or simplification of the traditional steps in a systematic review) compared to systematic review.

An out-dated rapid review conclusions cannot provide timely evidence for decision-makers, hence is meaningless. Mentioning the poor timeliness problem as an limitation does not mean a higher tolerance of this essential issue. 

This study only "sacrificed" the rigor of methodology (e.g., small sample size) using a rapid review standard rather than a systematic review standard,  but at the meantime, failed to meet the timeliness requirement. 

Hence, I believe the current version is not acceptable for publication.
